# Predictors of criticism and emotional over-involvement in relatives of early psychosis patients

Lídia Hinojosa-Marqués[1], Tecelli Domínguez-Martínez[2], Thomas R. Kwapil[3], Neus Barrantes-Vidal[1,4,5]*

**1** Departament de Psicologia Clínica i de la Salut, Universitat Autònoma de Barcelona, Barcelona, Spain, **2** Centro de Investigación en Salud Mental Global, Dirección de Investigaciones Epidemiológicas y Psicosociales, Instituto Nacional de Psiquiatría 'Ramón de la Fuente Muñiz', Mexico City, Mexico, **3** Department of Psychology, University of Illinois at Urbana-Champaign, Champaign, IL, United States of America, **4** Sant Pere Claver- Fundació Sanitària, Barcelona, Spain, **5** Centre for Biomedical Research Network on Mental Health (CIBERSAM), Instituto de Salud Carlos III, Barcelona, Spain

* neus.barrantes@uab.cat

## Abstract

Mechanisms underlying the manifestation of relatives' expressed emotion (EE) in the early stages of psychosis are still not properly understood. The present study aimed to examine whether relatives' psychological distress and subjective appraisals of the illness predicted EE dimensions over-and-above patients' poor clinical and functional status. Baseline patient-related variables and relatives attributes comprising criticism, emotional over-involvement (EOI), psychological distress, and illness attributions were assessed in 91 early psychosis patients and their respective relatives. Relatives were reassessed regarding EE dimensions at a 6-month follow-up. Relatives' psychological distress and illness attributions predicted criticism and EOI over-and-above patients' illness characteristics at both time points. Relatives' increased levels of anxiety, attributions of blame toward the patients, an emotional negative representation about the disorder, and decreased levels of self-blame attributions predicted EE-criticism at baseline. Relatives' anxiety and negative emotional representation of the disorder were the only significant predictors of EE-criticism at follow-up, whereas anxiety, attributions of control by the relative and an emotional negative representation about the disorder predicted EE-EOI both at baseline and follow-up assessments. Understanding the components that comprise and maintain EE attitudes should guide early psychosis caregivers in family interventions, enhancing proper management of psychological distress and reduction of negative appraisals about the illness. The prevention of high-EE attitudes over time in a sensitive period such as early psychosis might be critical in shaping the health of caregivers and the outcome of the affected relatives.

**Data Availability Statement:** The authors of the present study confirm that some access restrictions apply to the data underlying the

findings. The consent form that participants signed before participating in the study, approved by the Ethics Committee of the Unió Catalana d'Hospitals (Comitè d'Ètica d'Investigació Clínica (CEIC); number 09-40) and by the Ethics Committee of the Universitat Autònoma de Barcelona (Comissió d'Ètica en l'Experimentació Animal i Humana (CEEAH); number 2679) imposes restrictions for making the data publicly available. Participants agreed for all the data collected to be available to the members of the research group Person-Environment Interaction in Psychopathology led by Prof. Neus Barrantes-Vidal (Address: Departament de Psicologia Clínica i de la Salut, Facultat de Psicologia, Edifici B, Universitat Autònoma de Barcelona, 08193 Cerdanyola del Vallès, Spain; telephone: +34 93 5813864; email: neus. barrantes@uab.cat). Data available on request. Requests should be addressed to the contact details provided above or to the Ethics Committee of the Universitat Autònoma de Barcelona (Comissió d'Ètica en l'Experimentació Animal i Humana, Address: Plaça Acadèmica, Rectorat, Edifici A, Universitat Autònoma de Barcelona, 08193 Cerdanyola del Vallès, Spain; telephone: +34 93 5813578; email: oh.ceea@uab.cat).

**Funding:** Authors are supported by the Spanish Ministerio de Economía y Competitividad (PSI2017-87512-C2-01) and the Comissionat per a Universitats i Recerca of Generalitat de Catalunya (2017SGR1612). N. Barrantes-Vidal is supported by the Institució Catalana de Recerca i Estudis Avançats (ICREA) Academia Award and the Centro de Investigación Biomédica en Red de Salud Mental (CIBERSAM), Instituto de Salud Carlos III, Barcelona, Spain. The funders had no role in study design, data collection and analysis, decision to publish, or preparation of the manuscript.

**Competing interests:** The authors have declared that no competing interests exist.

## Introduction

Epidemiological research indicates that psychosocial factors impact the risk, progression and outcome of psychosis liability. In fact, some metanalyses have found odds ratios for macroenvironmental factors (e.g., urbanicity) that are similar to those attributed to genetic factors [1,2]. In the context of microenvironmental factors, one of the most significant factors in psychosocial research in psychosis has been expressed emotion (EE) [3], a measure of the family environment used to describe relatives' attitudes toward an ill family member. High-EE attitudes, particularly criticism and emotional over-involvement (EOI), are considered the strongest psychosocial predictors of relapse in schizophrenia [4]. Given the heightened stress-sensitivity associated to psychosis liability [5], the impact of high negative emotionality and distress in the family environment appears particularly relevant. However, it is important to note that the EE construct does not aim to blame families for contributing unidirectionally to the patient's clinical worsening. Instead, EE is best regarded as the product of a negative dynamic interaction between patients and their families that can play a central role in the course of the patient's disorder [6,7].

Although over the last 50 years most studies on EE have involved patients with schizophrenia, recent research has focused on the study of EE in relation to schizotypy [8], and the early course of psychosis [9,10]. Preliminary studies indicate that high-EE is already present in over half of the relatives of persons with first-episode psychosis (FEP) [11,12], and is even present in relatives of at-risk mental state (ARMS) [13,14]. Importantly, early psychosis caregivers often report a heightened risk of psychological distress and more negative caregiving appraisals compared to family members of patients with chronic schizophrenia [15]. Besides, recent findings indicated that relatives' high levels of depression and anxiety are associated with high levels of EE at these early stages [16–18]. Since the presence of EE is associated with multiple negative outcomes for relatives and patients, it is crucial to examine the mechanisms underlying the development of EE in the early stages of psychosis, when most of the changes are emerging and it is still possible to examine these factors without the bias created by chronic psychosis and relatives' long-term burden.

Although patients' poor clinical and functional status have been related with increased relatives' EE in several early psychosis studies [10,19,20], other studies have suggested that patients' symptoms/functioning have limited or no impact upon relatives' EE [21–23]. The differences in the results among the above cited studies leaves unanswered the question as to the extent that EE is a reaction to the severity of the relative's psychotic disorder [12].

The attributional model of EE suggests that relatives' beliefs about the nature of the illness explains relatives' emotional attitudes better than the unidirectional reactivity of relatives to patients' illness characteristics [24]. Specifically, critical relatives are more prone to blame patients for their behaviors and perceive symptoms as controllable by patients rather than illness driven; as a result, they react with criticism to reduce undesired behaviors. In contrast, overinvolved relatives may believe that they have contributed in some way to the patients' problems, so they often report high levels of self- control or self-blame attributions [25–27]. Some studies have empirically supported the attributional model in early psychosis. For example, critical relatives of FEP patients tend to believe that symptoms are within the patients' control [28,29], and attributions of blame toward the patient have been shown to predict relatives' criticism in the early stages of psychosis [16].

A different explanatory model of EE proposes that high-EE attitudes may represent a maladaptive attempt to cope with the stress of caring for an impaired ill relative; thus, EE behaviors could be conceived as a coping strategy used to reduce the perceived stress related to the caregiving role [21]. Accordingly, greater levels of psychological distress in caregivers seem to

be related with increased levels of EE [16–18]. Converging evidence suggests that EOI is more related to distress than criticism [17,18,21], although criticism has also been linked to psychological distress in early psychosis caregivers [30]. Of note, some cross-sectional studies have pointed that relatives' distress is an important predictor of criticism in early psychosis relatives [16,31].

Despite research aimed at disentangling the multiple correlates of EE, little is known about the developmental precursors of EE in early psychosis. To date, only a few longitudinal studies have explored potential predictors of criticism and EOI in relatives of FEP patients [21, 32]. Besides, the possible contribution of patients' clinical and functional characteristics on EE in conjunction with relatives' psychological factors has been scarcely investigated at these early stages [e.g., 21, 23]. Therefore, understanding the specific and common underlying factors of relatives' criticism and EOI in the early stages of psychosis should improve the design of early family interventions, thereby enhancing the prognosis for both patients and their relatives.

The current study aims to examine the association of relatives' EE with other family factors and patient's clinical/functional status in incipient psychosis. It expands previous preliminary cross-sectional reports [16] by using an extended sample and a longitudinal design to test whether patient's clinical/functional status and relatives' psychological factors prospectively predict relatives' EE at 6 months. The specific goals were to explore: (1) the association of patients' clinical and functional status, as well as relatives' psychological distress and illness attributions, with relatives' EE dimensions (criticism and EOI) at baseline and at the 6-month follow-up; and (2) whether relatives' psychological distress and illness attributions at the initial assessment predicted relatives' EE dimensions both at baseline and follow-up assessments over-and-above patients' baseline clinical and functional status.

Unlike previous preliminary cross-sectional studies, this study seeks to test a more comprehensive predictive model of EE in early psychosis by including *both* patients' illness-related variables and relatives' psychological factors. Moreover, the use of a longitudinal design allowed us to explore the assumption that relatives' psychological factors predict EE levels (over-and-above patients' clinical and functional features) across time at both subclinical and onset stages of psychosis.

Based on previous suggestions of the EE literature, we hypothesized that relatives' baseline psychological distress and negative illness attributions would predict relatives' EE dimensions over-and-above patients' baseline clinical and functional variables at both time points. Based on the attributional model of EE and on previous findings, we also expected common as well as distinctive predictors of criticism and EOI at both baseline and follow-up assessments: (a) Both criticism and EOI would be predicted by relatives' psychological distress (anxiety and depression); (b) Beliefs of self-blame, self-control, and emotional negative representation of the disorder would predict relatives' EOI, whereas beliefs of control and blame toward the patient would predict relatives' criticism.

## Materials and methods

### Participants and procedure

Ninety-one early psychosis patients (55 ARMS and 36 FEP) and their respective relatives were initially recruited in the present study. Of these, 46 family members (33 of ARMS and 13 of FEP) completed the 6-month follow-up assessment. Attrition results from the following reasons: 15 pilot participants completed the baseline assessment protocol but not the 6-months follow-up, 10 refused to participate, 10 did not complete the follow-up assessment, and 10 left the study because of patient withdrawal from the clinic or the study. Please note that patients' data refers only to the baseline assessment.

Relatives were those who had most contact and/or the most significant relationship with the patient. Patients had to meet ARMS criteria as assessed by the Comprehensive Assessment of At-Risk Mental States (CAARMS) [33] and/or the Schizophrenia Proneness Instrument Adult-Version (SPI-A) [34]. FEP patients met DSM-IV-TR criteria [35] for any psychotic disorder or affective disorder with psychotic symptoms as established by the Structured Clinical Interview for DSM-IV (SCID-I) [36] and presented a first-episode of psychosis within the past two years. Mean duration of illness was 14 months (SD = 9.8), although 4 patients slightly exceed the 24-month period (range 1 to 29 months) and 2 reached a length of 33 and 34 months. Exclusion criteria for patients were (a) evidence of organically based psychosis, (b) any previous psychotic episode that involved pharmacotherapy, and (c) intellectual disability.

Seventy-eight relatives and forty-four patient-relative dyads were included in previously published preliminary studies exploring cross-sectional associations of the EE dimensions with patients' clinical/functional status [19] and differences on the relationship of EE with distress and illness attributions in early psychosis stages [16].

All participants were recruited within the Sant Pere Claver-Early Psychosis Program conducted in Barcelona (Spain) [37] and provided written consent to participate. The project was developed in accordance with the Code of Ethics of the World Medical Association (Declaration of Helsinki). Ethical approval was granted by the Ethics Committee of the Unió Catalana d'Hospitals (Comitè d'Ètica d'Investigació Clínica (CEIC); number 09–40) and by the Ethics Committee of the Universitat Autònoma de Barcelona (Comissió d'Ètica en l'Experimentació Animal i Humana (CEEAH); number 2679). All the interviews were conducted by experienced clinical psychologists. The time gap between patients and relatives' assessments ranged from 3 to 15 days.

## Measures

**Relatives' measure at baseline and 6-month follow-up.**   EE was measured with the Family Questionnaire (FQ) [38] which consists of 20 items equally distributed in two subscales (criticism and EOI) scored on a 4-point Likert scale ranging from 'never/very rarely' to 'very often'.

**Patients and relatives' measures at baseline.**   ARMS and FEP patients' clinical status at baseline was rated with the Positive and Negative Syndrome Scale (PANSS) [39]. Patients' current functional status was measured with the short version of the Social Functioning Scale (SFS) [40], a self-reported measure that assesses multiple facets of social adjustment.

Relatives' distress was measured with the Depression and Anxiety subscales of the Symptom Checklist (SCL-90-R) [41], a self-report inventory intended to measure symptom intensity on a 5-point Likert scale. The Illness Perceptions Questionnaire for Schizophrenia-Relatives version (IPQS-R) [42] was used to measure relatives' beliefs about the disorder. Each item is rated from 1 'strongly disagree' to 5 'strongly agree'. For the purposes of this study, we used the following subscales: personal control-patient and personal control-relative (control over the disorder), personal blame-patient and personal blame relative (blame toward the patient or self-blame about the disorder), and emotional representation of the illness (negative emotions about the disorder including sense of fear, frustration, anger, worry).

## Statistical analysis

Pearson correlations were used to analyze the associations between each of the baseline predictors and relatives' outcome variables (EE-criticism and EE-EOI) at baseline and at the 6-month follow-up. Hierarchical regression analyses were computed to predict relatives' EE-criticism and EE-EOI at baseline and 6-month assessments using patients' and relatives' baseline predictors. The goal of the regression analyses was to test the extent to which relatives'

baseline predictors accounted for variance in relatives' EE-criticism and EE-EOI (at baseline and at the 6-month follow-up) over-and-above patients' baseline symptom severity and social functioning. The following steps were entered in all regression analyses. The PANNS total score was entered at step 1 to examine the variance accounted for by patients' symptom severity. Patients' SFS score was entered at step 2 to examine the variance accounted for by patients' social functioning. Relatives' Depression and Anxiety SCL-90-R subscales were entered at step 3 to examine the variance accounted for by relatives' psychological distress. The five IPQ-S subscales were entered as a block at step 4 to examine the variance accounted for by relatives' illness attributions.

## Results

### Sample characteristics

At baseline, relatives were mainly female (74.7%), particularly patients' mothers (71.4%), with the remaining caregivers being fathers (17.6%), siblings (5.5%), partners (4.4%) or adoptive parents (1.1%). Mean age of the relatives was 51.7 years old (S.D = 10.0). Patients were predominantly male (69.2%) and lived with their families (91.2%). The mean age of the patients was 22.4 years old (SD = 4.9). Over half of them (58.3%) were studying or had a job, 36.3% were unemployed/unoccupied, and 5.5% had a sick leave.

Of the 46 relatives that completed the 6-month follow-up assessment, most were also females (80.4%) and mothers who lived with the patient (89.1%). The mean age of the relatives at follow-up was 51.9 years old (S.D = 8.7).

Descriptive baseline data for all relatives' and patients' measures are presented in Table 1. The mean of the EE dimensions at follow-up was 20.52 (SD = 6.15) and 22.74 (SD = 5.48) for EE-criticism and EE- EOI, respectively. Dependent *t*-tests revealed that, on average, the levels of EE-criticism (*t* (45) = -0.670, *p* = 0.51) and EE-EOI (*t* (45) = 0.834, *p* = 0.41) did not differ significantly between baseline and follow-up assessments.

### Associations of EE with patients and relatives' variables

Results of the association of patients' and relatives' baseline measures with relatives' levels of criticism and EOI at both baseline and follow-up are provided in Table 2. Regarding the cross-sectional associations, only patients' social functioning was significantly associated with relatives' EE-criticism, such that worse social functioning in patients at baseline was related to relatives' increased levels of criticism. No associations were found between patients' baseline clinical status and relatives' baseline EE indices. Furthermore, no longitudinal associations were found between patients' variables at baseline and relatives' EE at follow-up.

As for the associations between relatives' baseline psychological distress and relatives' EE dimensions, both relatives' baseline levels of anxiety and depression were strongly associated with relatives' EE dimensions at both time points. Moreover, relatives' baseline attributions of blame toward the patient were significantly related with relatives' EE-criticism at baseline, whereas relatives' baseline attributions of control by the relative were significantly associated with relatives' EE-EOI at baseline. Finally, relatives' emotional negative representation about the disorder at baseline showed significant associations with both EE dimensions at both time points. Neither attributions of control toward the patient nor self-blame attributions were associated with EE indices.

### Predictors of EE at baseline

Hierarchical regressions showed that neither patients' baseline clinical status nor baseline psychosocial functioning accounted for significant variance in the prediction of baseline EE

**Table 1. Descriptive baseline data of early psychosis patients and their respective relatives (n = 91).**

|  | α | Possible score range | Observed Score Range | Mean (SD) |
|---|---|---|---|---|
| *Patients* |  |  |  |  |
| **Clinical status (PANSS)** |  |  |  |  |
| Positive symptoms | - | 7–49 | 7–24 | 12.74 (3.40) |
| Negative symptoms | - | 7–49 | 7–34 | 17.40 (5.80) |
| General symptoms | - | 16–112 | 18–66 | 32.93 (7.51) |
| **Functional status (SFS)** |  |  |  |  |
| Social Functioning | 0.76 | 0–43 | 2–36 | 21.19 (6.12) |
| *Relatives* |  |  |  |  |
| **Expressed Emotion (FQ)** |  |  |  |  |
| Criticism | 0.86 | 10–40 | 10–36 | 20.86 (6.15) |
| EOI | 0.82 | 10–40 | 11–36 | 23.98 (5.82) |
| **Distress (SCL-90-R)** |  |  |  |  |
| Anxiety | 0.90 | 0–40 | 0–34 | 7.13 (7.41) |
| Depression | 0.91 | 0–52 | 0–39 | 14.73 (10.62) |
| **Illness Attributions (IPQS-R)** |  |  |  |  |
| Personal control-Patient | 0.63 | 4–20 | 8–20 | 14.69 (2.58) |
| Personal control-Relative | 0.61 | 4–20 | 6–20 | 13.45 (2.68) |
| Personal blame-Patient | 0.81 | 3–15 | 3–15 | 9.93 (2.94) |
| Personal blame-Relative | 0.83 | 3–15 | 3–15 | 7.45 (2.76) |
| Emotional representation | 0.81 | 9–45 | 10–41 | 27.76 (7.10) |

SD, Standard Deviation; PANSS, Positive and Negative Syndrome Scale; SFS, Social Functioning Scale; FQ, Family Questionnaire; EOI, Emotional Over-Involvement; SCL-90-R, Symptom Checkllist-90-Revised; IPQS-R, Illness Perception Questionnaire for Schizophrenia-Relatives' version.

indices (see Table 3). Relatives' baseline data revealed that high levels of anxiety (but not depression), attributions of blame toward the patient, emotional negative representation of the disorder as well as low scores on self-blame attributions significantly accounted for variance in relatives' EE-criticism at baseline. In addition, high levels of anxiety (but not depression), attributions of control by the relative and emotional negative representation about the disorder significantly accounted for variance in relatives' EE-EOI at baseline. However, ratings of relatives' depression and anxiety were highly collinear, and they had comparable zero-order associations with EE.

## Baseline predictors of EE at follow-up

Hierarchical regressions indicated that neither patients' baseline clinical status nor baseline psychosocial functioning accounted for significant variance in the prediction of follow-up EE indices (see Table 4). Relatives' high levels of anxiety (but not depression) and emotional negative representation about the disorder at baseline significantly accounted for variance in relatives' EE-criticism at follow-up. For EE-EOI, relatives' high levels of anxiety (but not depression), attributions of control by the relative and emotional negative representation about the disorder at baseline significantly accounted for variance at follow-up.

## Discussion

This study examined the association of relatives' EE with psychological distress and illness attributions as well as with patients' clinical features and functioning at baseline and at the 6-month follow-up. Furthermore, it was investigated whether relatives' psychological distress

**Table 2. Pearson correlations between baseline predictors (patient and relative factors) and relatives' EE dimensions at baseline and follow-up.**

|  | Baseline (n = 91) | | Follow-up (n = 46) | |
|---|---|---|---|---|
|  | EE-Criticism | EE-EOI | EE-Criticism | EE-EOI |
| *Patient factors* |  |  |  |  |
| **Clinical status (PANSS)** |  |  |  |  |
| Positive symptoms | 0.14 | 0.09 | 0.28 | 0.12 |
| Negative symptoms | 0.05 | 0.08 | -0.08 | 0.06 |
| General symptoms | 0.15 | 0.11 | 0.25 | 0.02 |
| PANSS total score | 0.13 | 0.12 | 0.18 | 0.07 |
| **Functional status (SFS)** |  |  |  |  |
| Social Functioning | -0.21* | -0.03 | -0.12 | -0.10 |
| *Relative factors* |  |  |  |  |
| **Distress (SCL-90-R)** |  |  |  |  |
| Anxiety | *0.60*\*\*\* | *0.51*\*\*\* | *0.61*\*\*\* | *0.62*\*\*\* |
| Depression | *0.59*\*\*\* | *0.50*\*\*\* | *0.63*\*\*\* | *0.62*\*\*\* |
| **Illness Attributions (IPQS-R)** |  |  |  |  |
| Personal control-Patient | 0.13 | 0.02 | 0.28 | 0.06 |
| Personal control-Relative | -0.02 | 0.21* | 0.09 | 0.29 |
| Personal blame-Patient | 0.27** | 0.07 | 0.27 | 0.23 |
| Personal blame-Relative | 0.04 | 0.09 | 0.10 | 0.27 |
| Emotional representation | **0.44**\*\*\* | *0.61*\*\*\* | *0.52*\*\*\* | *0.56*\*\*\* |

*p<0.05

**p≤ 0.01

*** p<0.001.

EOI, Emotional Over-Involvement; PANSS, Positive and Negative Syndrome Scale; SFS, Social Functioning Scale; SCL-90-R, Symptom Checkllist-90-Revised; IPQS-R, Illness Perception Questionnaire for Schizophrenia-Relatives' version.

Medium effect sizes (r ≥ 0.30) in bold, large effect sizes (r ≥ 0.50) in bold and italics.

**Table 3. Predictors of EE at baseline (N = 91 patients and their respective relatives).**

|  |  | EE-Criticism (at baseline) | | EE-EOI (at baseline) | |
|---|---|---|---|---|---|
| Step | Baseline Predictors | β | p | β | p |
| 1 | **Patients' clinical status (PANSS)** |  |  |  |  |
|  | PANSS total score | 0.131 | 0.22 | 0.117 | 0.27 |
| 2 | **Patients' functional status (SFS)** |  |  |  |  |
|  | Social Functioning | -0.188 | 0.10 | 0.012 | 0.92 |
| 3 | **Relatives' distress (SCL-90-R)** |  |  |  |  |
|  | Anxiety | **0.466** | **0.001** | **0.331** | **0.04** |
|  | Depression | 0.196 | 0.17 | 0.223 | 0.17 |
| 4 | **Relatives' illness attributions (IPQS-R)** |  |  |  |  |
|  | Personal control-Patient | 0.116 | 0.15 | 0.048 | 0.58 |
|  | Personal control-Relative | 0.001 | 0.99 | **0.207** | **0.02** |
|  | Personal blame-Patient | **0.240** | **0.008** | 0.024 | 0.80 |
|  | Personal blame-Relative | **-0.238** | **0.01** | -0.160 | 0.10 |
|  | Emotional representation | **0.250** | **0.004** | **0.465** | **0.000** |

EOI, Emotional Over-Involvement; PANSS, Positive and Negative Syndrome Scale; SFS, Social Functioning Scale; SCL-90-R, Symptom Checkllist-90-Revised; IPQS-R, Illness Perception Questionnaire for Schizophrenia-Relatives' version.

Significant at p<0.05 (two-tailed) are bolded.

**Table 4. Baseline predictors of EE at follow-up (N = 46 patients and their respective relatives).**

| Step | Baseline Predictors | EE-Criticism (at follow-up) | | EE-EOI (at follow-up) | |
|---|---|---|---|---|---|
| | | β | p | β | p |
| 1 | **Patients' clinical status (PANSS)** | | | | |
| | PANSS total score | 0.178 | 0.24 | 0.070 | 0.65 |
| 2 | **Patients' functional status (SFS)** | | | | |
| | Social Functioning | -0.080 | 0.61 | -0.082 | 0.60 |
| 3 | **Relatives' distress (SCL-90-R)** | | | | |
| | Anxiety | **0.449** | **0.02** | **0.410** | **0.04** |
| | Depression | 0.279 | 0.144 | 0.326 | 0.10 |
| 4 | **Relatives' illness attributions (IPQS-R)** | | | | |
| | Personal control-Patient | 0.185 | 0.13 | -0.018 | 0.88 |
| | Personal control-Relative | 0.024 | 0.84 | **0.233** | **0.04** |
| | Personal blame-Patient | 0.141 | 0.28 | 0.101 | 0.42 |
| | Personal blame-Relative | -0.089 | 0.50 | -0.005 | 0.97 |
| | Emotional representation | **0.278** | **0.04** | **0.346** | **0.01** |

EOI, Emotional Over-Involvement; PANSS, Positive and Negative Syndrome Scale; SFS, Social Functioning Scale; SCL-90-R, Symptom Checkllist-90-Revised; IPQS-R, Illness Perception Questionnaire for Schizophrenia Relatives' version.

Significant at $p < 0.05$ (two-tailed) are bolded.

and illness attributions predicted EE dimensions over-and-above patients' baseline clinical and functional status at both time points. Overall, findings support the attributional model of EE as specific and differential attributions predicted criticism and EOI. In addition, emotional distress appears as a critical factor in the expression and maintenance of both EE dimensions in the early stages of psychosis. Evidence confirms that early family-based interventions have great potential to reduce high-EE attitudes and improve patients' outcomes [43,44]. In general, family therapies aimed at reducing high-EE include training in family communication and problem solving in addition to psychoeducation [45,46]. However, relatives' own needs and the emotional impact of caregiving are still a neglected intervention area in the early stages of psychosis [44]. Thus, these findings highlight that early family interventions would benefit from providing relatives with proper psychological support. This could involve helping relatives to handle harming thoughts and emotions, facilitate emotional processing, and provide specific techniques to reduce negative appraisals and stress. This, in turn, might prevent relatives' high-EE over the psychotic process.

Consistent with some previous early psychosis studies [12,21,23], no associations were found between patients' clinical status and relatives' EE at baseline or follow-up. Of the patient-related variables, only patients' poor social functioning was slightly related to relatives' criticism at baseline, which is consistent with previous cross-sectional findings [19,23]. However, it is important to note that most early psychosis studies show inconsistent results regarding the relationship of EE with patients' symptoms/functioning. Such differences might be related to the variability of the samples analysed (as some studies use a combined sample of early psychosis participants whereas others only examined ARMS or FEP participants separately), and to the diversity of designs employed (e.g., comparison of ARMS with FEP, ARMS and FEP vs. health controls, ARMS and FEP vs. chronic psychosis). Besides, it is likely that the heterogeneity characterizing psychosis, even in the early stages of the disorder make it difficult to find a conclusive pattern regarding the associations between patients' illness-related characteristics and relatives' EE.

The significant associations of relatives' anxiety and depression symptoms with EE dimensions at both baseline and follow-up is consistent with previous cross-sectional findings [16–18, 30]. Although relatives' distress and EE may have a complex pattern of interactions, our findings suggest that early psychosis has a profound impact on relatives' emotional state that leads to significant levels of distress [47], which in turn may exacerbate EE attitudes.

Regarding the association between illness attributions and EE, findings indicated that attributions of blame toward the patient were significantly related with EE-criticism at baseline. This result support previous findings of cross-sectional studies [16,19] and lends further support to the attributional model [24], which states that relatives who believe that patients are guilty of their behaviors are more prone to manifest critical attitudes. Moreover, in agreement with Bolton et al. [26], it was found that relatives' self-control attributions were significantly associated with EE-EOI at baseline. Thus, relatives' who perceive themselves as responsible for the behaviors of their offspring are more likely to exhibit intrusive and/or self-sacrificing attitudes. Finally, strong relationships were observed between relatives' negative emotional representation of the disorder and both EE dimensions at baseline and follow-up. This further supports the assumption that EE attitudes may be driven by relatives' negative emotional responses to illness such as a sense of fear, frustration, anger or worry [21], which stands out as a crucial factor to consider in early family interventions.

The most relevant finding of this study is that relatives' levels of anxiety and several illness attributions accounted for significant variance over-and-above patient-related variables in the prediction of criticism and EOI both at baseline and the 6-month follow-up. These results confirm that, as in schizophrenia, relatives' emotional state, along with the cognitive representation of psychosis, are playing an important role in the emergence of emotional attitudes toward the patient. In particular, relatives' increased levels of anxiety, attributions of blame toward the patient, decreased levels of self-blame attributions and an emotional negative representation of the disorder predicted EE-criticism at baseline. These results are in accordance with Domínguez-Martínez et al. [16] thus suggesting that anxiety responses and attributions of blame toward the patient play an important role in predicting relatives' levels of criticism in early psychosis. These findings are also consistent the attributional model of schizophrenia [24], which posits that relatives who attribute responsibility or blame to the patients for their behaviors are more likely to exhibit critical attitudes towards them. The fact that decreased levels of self-blame attributions were predictive of relatives' criticism at baseline is not surprising given that critical relatives tend to place the blame or attribute the responsibility of illness behaviors mainly to patients' internal/personal factors [48], which could reduce feelings of guilt or responsibility for patients' problems. Finally, our results pointed out the role of relatives' negative emotional representation of patients' disorder as an important cognitive/affective predictor of EE-criticism at baseline.

Interestingly, in the regression analyses, the only predictors of relatives' EE-criticism at follow-up were relatives' anxiety levels and an emotional negative representation of the disorder. Attributions of blame toward the patient and/or low levels of self-blame attributions lost its predictive value at follow-up. This may reflect that relatives receive adequate information during treatment regarding the nature and causes of the behaviors of their ill relative, which helps them to better understand the cause of the problem and thus reduce their blaming attributions. Of note, emotional negative representation of illness, the only attribution that encompasses both cognitive and emotional components (e.g., sense of fear, frustration, anger, worry), predicted relatives' criticism both at baseline and follow-up assessments over-and-above all the predictors in the model. This suggests that negative emotional responses (i.e., anxiety) and cognitions that are clearly intertwined with emotions (i.e., emotional negative representation of the disorder) persistently promote the maintenance of EE-criticism, even when 'core'

cognitive attributions of blame no longer seem to have a relevant role. A similar pattern of results was observed for EE-EOI. Both relatives' anxiety levels and emotional negative representation of the disorder appeared as significant predictors of relatives' EOI at both assessments' points, which further indicates that the emergence of criticism and EOI attitudes stems from negative emotional responses and negative 'hot' cognitions. Overall, these findings indicate that emotional distress is a critical element for understanding the expression and maintenance of EE in the early phases of psychosis and suggests that therapeutic efforts should address negative affective processes in addition to the more generally implemented psychoeducational techniques that predominantly focus on cognitive restructuring.

Consistent with the results of Bolton et al. [26], and lending further support to the attributional model, EE-EOI was also predicted by relatives' self-control attributions both at baseline and follow-up, which may indicate that, unlike attributions of blame towards the patient, parental beliefs regarding their capacity to change their offspring's situation are more difficult to change as they tap on core aspects of the parental function (e.g., protection). Thus, overinvolved behaviors might be conceptualized as reflecting a hyperactivation of the parental system in a moment of crisis. Therefore, the malleability of this attribution and its impact on parental behaviors might require overcoming the acute crisis state characterizing early psychosis.

Consistent with previous studies, attributions of control toward the patient did not predict relatives' criticism either at baseline or follow-up [16,19]. These findings seem to suggest that attributions of blame toward the patient, rather than attributions of control, have a significant impact on relatives' criticism. It is thus attractive to speculate that attributions of blame toward the patient may be emotionally-driven in response to crisis situation defining the early stages of psychosis, but that in later and chronic stages they may turn into stronger beliefs of patient's personal control over the illness. Furthermore, contrary to our hypotheses, and in contrast with some schizophrenia studies [25,27], self-blame attributions did not predict relatives' EOI either at baseline or follow-up. It might be that self-blame attributions tend to become more evident over time when diagnoses are clearly established, as relatives might then wonder about their contribution in the development of illness.

The strengths of this study include the use of a longitudinal design, which allowed to test prospectively the assumption that relatives' attributions and distress predict EE levels across time in both subclinical and onset stages of psychosis. Besides, the present study examined a more comprehensive predictive model of EE in early psychosis by including both patients' illness-related variables and relatives' emotional distress and illness attributions. A limitation of this study is that the FQ does not contemplate the positive components of the EE construct (e.g., warmth). Considering that preliminary research emphasizes the protective effect of warmth in both chronic [49] and early psychosis samples [13,14, 50], further research should examine the psychological mechanisms underlying the expression of positive components of EE and its potential buffering effect of negative components on the course of early psychosis. Another limitation is that all measures administered to relatives (e.g., EE, attributions and distress) were assessed using self-reported scales, which raises concern about the contribution of shared method variance in the findings reported. The use of additional observational and/or interview ratings of these family constructs would strengthen the validity of these findings and should be pursued in future studies. Finally, it is important to note that content overlap among measures (and even constructs) of EE, anxiety and relatives' emotional representation of the patients' disorder may have partly contributed to the observed relationships. That is, some of the items of measures that assess different constructs present some overlap (i.e., they tap comparable content). A specific example is that both the EE-EOI subscale of the FQ [38] and the relatives' emotional representation of the disorder subscale of the IPQS-R [42] contained items assessing relatives' worry about the patient (e.g., an item of the EE-EOI subscale reads 'I'm

very worried about him/her', and an item of the relatives' emotional representation of the disorder subscale reads 'Their mental health problems do not worry me'). However, it is important to note that this measurement overlap probably reflects to some extent a certain degree of construct overlap. EE, anxiety and relatives' emotional representation of the disorder probably all partly draw on a general factor of relatives' emotional distress.

In conclusion, this study shows that relatives' psychological distress and negative illness attributions accounted for significant variance over-and-above patients' clinical and functional status in the prediction of EE attitudes both at baseline and 6-month follow-up assessments. This highlights the importance of considering how relatives' emotional states and the early formation of cognitive representations of psychosis can affect the attitudes they take towards the disorder at the at-risk and recent onset stages of psychosis. Moreover, these results underscore the need for early family interventions to go beyond educating relatives about psychosis [e.g., 51] to attend to their specific psychological needs. The findings of the present study suggest that providing relatives with stress management and problem-solving techniques [52] along with the use of reattribution techniques [24] might be beneficial to acquire a more balanced emotional and/or attributional stance. This, in turn, might help relatives to gain insight over their EE attitudes and manage them more effectively.

## Acknowledgments

The authors appreciate the support offered by the clinicians and all members of the staff of the Fundació Sanitària Sant Pere Claver who provided access to the families that participated in the study. Specially thanks to the patients and their respective relatives who consented to participate. We acknowledge Cristina Medina-Pradas and Nieves Guardia for their contribution in the data collection.

## Author Contributions

**Conceptualization:** Lídia Hinojosa-Marqués, Thomas R. Kwapil, Neus Barrantes-Vidal.

**Data curation:** Lídia Hinojosa-Marqués, Tecelli Domínguez-Martínez, Thomas R. Kwapil, Neus Barrantes-Vidal.

**Formal analysis:** Lídia Hinojosa-Marqués, Thomas R. Kwapil, Neus Barrantes-Vidal.

**Funding acquisition:** Neus Barrantes-Vidal.

**Investigation:** Lídia Hinojosa-Marqués, Tecelli Domínguez-Martínez, Neus Barrantes-Vidal.

**Methodology:** Thomas R. Kwapil, Neus Barrantes-Vidal.

**Project administration:** Neus Barrantes-Vidal.

**Resources:** Neus Barrantes-Vidal.

**Software:** Thomas R. Kwapil, Neus Barrantes-Vidal.

**Supervision:** Thomas R. Kwapil, Neus Barrantes-Vidal.

**Validation:** Thomas R. Kwapil.

**Visualization:** Lídia Hinojosa-Marqués, Tecelli Domínguez-Martínez, Neus Barrantes-Vidal.

**Writing – original draft:** Lídia Hinojosa-Marqués.

**Writing – review & editing:** Lídia Hinojosa-Marqués, Tecelli Domínguez-Martínez, Thomas R. Kwapil, Neus Barrantes-Vidal.

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
