## [Decision Letter · Decision Letter 0]

12 Feb 2020

PONE-D-19-32832

Predictors of criticism and emotional over-involvement in relatives of early psychosis patients

PLOS ONE

Dear Dr. Barrantes-Vidal,

Thank you for submitting your manuscript to PLOS ONE. After careful consideration, we feel that it has merit but does not fully meet PLOS ONE’s publication criteria as it currently stands. Therefore, we invite you to submit a revised version of the manuscript that addresses the points raised during the review process.

We would appreciate receiving your revised manuscript by Mar 28 2020 11:59PM. To enhance the reproducibility of your results, we recommend that if applicable you deposit your laboratory protocols in protocols.io, where a protocol can be assigned its own identifier (DOI) such that it can be cited independently in the future. For instructions see: http://journals.plos.org/plosone/s/submission-guidelines#loc-laboratory-protocols

We look forward to receiving your revised manuscript.

Kind regards,

Sarah Hope Lincoln

Academic Editor

PLOS ONE

Journal Requirements:

Reviewers' comments:

Reviewer's Responses to Questions

**Comments to the Author**

1. Is the manuscript technically sound, and do the data support the conclusions?

Reviewer #1: Partly

Reviewer #2: Yes

2. Has the statistical analysis been performed appropriately and rigorously? 

Reviewer #1: Yes

Reviewer #2: Yes

3. Have the authors made all data underlying the findings in their manuscript fully available?

Reviewer #1: Yes

Reviewer #2: Yes

4. Is the manuscript presented in an intelligible fashion and written in standard English?

Reviewer #1: Yes

Reviewer #2: Yes

5. Review Comments to the Author

Reviewer #1: The authors frame this study as examining the “mechanisms underlying the ontogenesis of expressed emotion (EE) in the early stages of psychosis.” This is clearly an important area of research as the findings have implications for intervention, particularly family interventions. The authors examine the role carers’ psychological distress and subjective appraisals of the illness relate to their level of criticism and emotional overinvolvement, two important predictors of illness course. The major strength of this study is the prospective design. They assessed not only the interrelation of the variables of interest at baseline but also at a six month follow-up. The authors found that family carers’ psychological distress (anxiety) and “negative emotional representation of the disorder” at baseline predicted 6 months later the carers’ level of criticism directed at the person with early psychosis. In addition, they observed that together a set of baseline indicators, specifically the relatives’ level of anxiety, their perceptions of their ability to control their loved ones mental health problems, and again the “negative emotional representation of the disorder” predicted their level of emotional overinvolvement at follow-up. The importance of the carers’ psychological distress in predicting both criticism and emotional overinvolvement calls for attending to caregivers’ distress in the early stages of the illness course of psychotic disorders. This strikes me as an important point.

Despite the strengths of the paper and its potential contribution, my enthusiasm for the paper is dampened by a number of weaknesses. Except for the symptomatology and social functioning scales, all the measures are based on the carers’ self report. This becomes a problem as there is some overlap between the measures, particularly family carers’ level of anxiety, “negative emotional representation of the disorder”, and emotional overinvolvement. The SCL-90R is a well-validated measure of anxiety symptoms. A look at the Illness perception questionnaire for schizophrenia (Lobban et al., 2005) reveals that of the nine items that assess “Emotional representation”, three include anxiety related items: “Their mental health problems do not worry me, their mental health problems make me feel anxious, and their mental health problems make me feel afraid.” A look at the 10 items to assess emotional overinvolvement (Wiedemann et al., 2002) also reveals anxiety related items. “I’m very worried about him/her” strikes me as the most obvious. Another is “I often think about what is to become of him/her,” although this is less direct. Clearly the measures of emotional representation and EOI include a number of what appear to be nonanxiety related items (e.g., the EOI scale has a number of items measuring self-sacrifice). The measure of emotional representation also overlaps with the measure of criticism. Two items stand out: “Their mental health problems make me feel angry” and “I get very frustrated by their mental health problems.” One item of the measure of criticism is “I’m often angry with him/her.” Two others seem related to frustration, those that refer to annoyance and irritation. My concern is that at least some of the observed associations may be a reflection of common method variance, not just regarding the use of self-report measures, but also regarding some of the measures content.

I found the speculation about the shift of attributions of blame to control over the illness course as problematic. First of all, the measures of controllability attributions in some of the prior studies of schizophrenia (e.g., Weisman et al, 1998, and Brewin et al., 1991) are quite different (based on coders’ ratings of statements generated from the Camberwell Family Interview). The authors would be on stronger grounds if the presuimed difference was observed across the illness course using similar methods. Second, I find the cognition-emotion distinction muddled in this speculation, and to some extent throughout the paper. Weiner’s attribution of responsibility model, which underlies some of the attribution-EE literature, makes a clear distinction between the appraiser’s cognitive construal and the subsequent affective reaction. To refer to emotional negative representation of illness as “the only attribution that encompasses both cognitive and emotional components (e.g., sense of fear, frustration, anger, worry) blurs the conceptual distinction between cognition and affect.

On a few occasions the authors point out that their findings are consistent with their prior work (Dominguez-Martinez et al., 2014; Dominguez-Martinez et al., 2017) and other’s work as well. For example they wrote on p. 13 that “The significant associations of relatives’ anxiety and depression symptoms with EE dimensions at both baseline and follow-up is consistent with previous cross-sectional findings [14-16, 18]. Also on page 14 they noted, “Regarding the association between illness attributions and EE, findings indicated that attributions of blame toward the patient were significantly related with EE-criticism at baseline. This result support (sic) previous findings of cross-sectional studies [14,17] and lends further support to the attributional model [22], which states that relatives who believe that patients are guilty of their behaviors are more prone to manifest critical attitudes. What concerns me is that earlier in the paper they report that the findings of the previous two studies were based on 78 relatives and 44 patient-relative dyads from the current study. That turns out to be 86% of the relatives (78 of 91) and 95% of the dyads (44 of 46). To refer to the findings from previous studies based on nearly the same sample as being supportive of the current findings is potentially misleading.

The last major point is that the authors refer to the ontogenesis and origin of expressed emotion in the first sentence of the abstract and the last sentence in the first paragraph of the discussion section. Indeed the researchers are studying these processes early in the illness course. The authors provide no evidence, however, that they captured the actual beginning of the caregivers’ attitudes and emotional reactions that reflect expressed emotion. They might want to be more circumspect in describing the study and interpreting their findings as an examination of the period of early psychosis, which they do throughout most of the paper.

Minor concerns

It would be helpful to briefly mention the ARMS criteria and the criteria they used to identify first episode psychosis. For example, was there any consideration of time since illness onset as seen in some first episode studies (Srihari et al., 2014)?

In assessing the carers’ level of anxiety, the authors report using SCL-90 R, yet they cite a much earlier paper that refers to the SCL-90 not the SCL-90R.

One limitation that the authors point out is that they did not consider the role of carers’ positive affect. The authors might consider other research that points out that carers’ supportive stance is associated with positive clinical outcomes for early stages of psychosis (Lee et al., 2014).

Reviewer #2: The present study is examining potential predictors of expressed emotion (EE), specifically emotional over involvement (EOI) and criticism in relatives of early psychosis patients. Overall, this is an interesting study and has important implications for early psychosis interventions. The manuscript is well-written, the justification for the study is clear, and is overall an interesting contribution to the growing field of understanding how family dynamics may interact with an existing vulnerability to severe mental illness. Nevertheless, the methodological concerns stated below should be emphasized. In addition, a few suggestions/comments:

Introduction

• In the first paragraph, please specify or give examples of the "macroenvironmental factors" referenced. Aside from EE, which is the focus of this paper, what else might be at play? Or is this primarily what the authors mean by macroenvironment?

• At the end of the first paragraph of the introduction, and throughout the paper, be careful not to imply blame on the family system for exacerbating symptoms or causing relapse, but rather clarifying that the family can play an important role in the course of the patient's illness.

• In the introduction, family therapy is briefly mentioned, and a specific study (O'Brien et al) is referenced at the very end of the discussion. Although the focus of this paper is not intervention-based, there are important intervention implications. It would be helpful to spend 1-2 lines (earlier than the last paragraph of the discussion) speaking more specifically about how EE is currently addressed in family therapy and how it could be improved with the findings from the current study.

Methods

• It was surprising that there was no inclusion of positive emotion or protective family factors (e.g. warmth, constructive communication) to use as a differential comparison and no mention of these factors until almost the very end of the discussion. These seems like a major limitation of the study (as is briefly mentioned) and should perhaps be addressed earlier on as to why they were not included. For example, I would be interested to know not only whether higher levels of distress and attributions of illness control predict EE but whether, differentially, lower levels of warmth and constructive communication also predict this. And, conversely, how levels of distress and illness attributions differentially predict factors like warmth and constructive communication. This double dissociation would be crucial to improving the impact of this finding.

• Why weren't demographic variables such as age and sex included as covariates in the hierarchical regression models? There seems to be some variance at least in age and it would be interesting and important to see whether this interacts with the variables of interest to predict EE.

Results

• In table 1, it would be helpful and more clear for the reader to include 6 month observations side-by-side with the baseline assessments for comparison, in addition to t-tests to compare measures across the two time points.

General

It would be helpful for the reader throughout the manuscript to make it more clear which measures refer to the patient, and which refer to the relatives. For example, in the abstract, the first line refers to the "ontogenesis of expressed emotion in the early stages of psychosis" but it could be made more clear that this is EE in relatives, not patients.

6. PLOS authors have the option to publish the peer review history of their article (what does this mean?). If published, this will include your full peer review and any attached files.

Reviewer #1: No

Reviewer #2: No

---

## [Author Response · Author response to Decision Letter 0]

30 Mar 2020

Summary of revisions to the manuscript, Predictors of criticism and emotional over-involvement in relatives of early psychosis patients (PONE-D-19-32832)

Response to Academic Editor 

Thank you for submitting your manuscript to PLOS ONE. After careful consideration, we feel that it has merit but does not fully meet PLOS ONE’s publication criteria as it currently stands. Therefore, we invite you to submit a revised version of the manuscript that addresses the points raised during the review process.

Thank you for the constructive reviews of our manuscript, “Predictors of criticism and emotional over-involvement in relatives of early psychosis patients”. We are delighted to have the opportunity to submit a revised manuscript, and we have revised the manuscript according to the Reviewers’ useful recommendations and provide a detailed summary of the revisions below. 

Response to Reviewer 1 

1. The authors frame this study as examining the “mechanisms underlying the ontogenesis of expressed emotion (EE) in the early stages of psychosis.” This is clearly an important area of research as the findings have implications for intervention, particularly family interventions. The authors examine the role carers’ psychological distress and subjective appraisals of the illness relate to their level of criticism and emotional overinvolvement, two important predictors of illness course. The major strength of this study is the prospective design. They assessed not only the interrelation of the variables of interest at baseline but also at a six-month follow-up. The authors found that family carers’ psychological distress (anxiety) and “negative emotional representation of the disorder” at baseline predicted 6 months later the carers’ level of criticism directed at the person with early psychosis. In addition, they observed that together a set of baseline indicators, specifically the relatives’ level of anxiety, their perceptions of their ability to control their loved ones mental health problems, and again the “negative emotional representation of the disorder” predicted their level of emotional overinvolvement at follow-up. The importance of the carers’ psychological distress in predicting both criticism and emotional overinvolvement calls for attending to caregivers’ distress in the early stages of the illness course of psychotic disorders. This strikes me as an important point.

We thank the Reviewer for the positive feedback on the study and manuscript. 

2. Despite the strengths of the paper and its potential contribution, my enthusiasm for the paper is dampened by a number of weaknesses. Except for the symptomatology and social functioning scales, all the measures are based on the carers’ self-report. This becomes a problem as there is some overlap between the measures, particularly family carers’ level of anxiety, “negative emotional representation of the disorder”, and emotional overinvolvement. The SCL-90R is a well-validated measure of anxiety symptoms. A look at the Illness perception questionnaire for schizophrenia (Lobban et al., 2005) reveals that of the nine items that assess “Emotional representation”, three include anxiety related items: “Their mental health problems do not worry me, their mental health problems make me feel anxious, and their mental health problems make me feel afraid.” A look at the 10 items to assess emotional overinvolvement (Wiedemann et al., 2002) also reveals anxiety related items. “I’m very worried about him/her” strikes me as the most obvious. Another is “I often think about what is to become of him/her,” although this is less direct. Clearly the measures of emotional representation and EOI include a number of what appear to be nonanxiety related items (e.g., the EOI scale has a number of items measuring self-sacrifice). The measure of emotional representation also overlaps with the measure of criticism. Two items stand out: “Their mental health problems make me feel angry” and “I get very frustrated by their mental health problems.” One item of the measure of criticism is “I’m often angry with him/her.” Two others seem related to frustration, those that refer to annoyance and irritation. My concern is that at least some of the observed associations may be a reflection of common method variance, not just regarding the use of self-report measures, but also regarding some of the measures content.

The reviewer points out very relevant issues. We completely concur with the Reviewer that findings may partly reflect shared method variance given that all measures administered to relatives had a self-report format. Also, content overlap among measures may have partly contributed to the observed relationships—and it could be argued that the boundaries among these constructs are themselves a challenging issue. Thus, we now make reference to these points in our discussion of the limitations of the study (page 17, lines 501 to 516) as follows: 

“Another limitation is that all measures administered to relatives (e.g., EE, attributions and distress) were assessed using self-reported scales, which raises concern about the contribution of shared method variance in the findings reported. The use of additional observational and/or interview ratings of these family constructs would strengthen the validity of these findings and should be pursued in future studies. Finally, it is important to note that content overlap among measures (and even constructs) of EE, anxiety and relatives’ emotional representation of the patients’ disorder may have partly contributed to the observed relationships. That is, some of the items of measures that assess different constructs present some overlap (i.e., they tap comparable content). A specific example is that both the EE-EOI subscale of the FQ [38] and the relatives’ emotional representation of the disorder subscale of the IPQS-R [42] contained items assessing relatives’ worry about the patient (e.g., an item of the EE-EOI subscale reads ‘I’m very worried about him/her’, and an item of the relatives’ emotional representation of the disorder subscale reads ‘Their mental health problems do not worry me’). However, it is important to note that this measurement overlap probably reflects to some extent a certain degree of construct overlap. EE, anxiety and relatives’ emotional representation of the disorder probably all partly draw on a general factor of relatives’ emotional distress.” 

3. I found the speculation about the shift of attributions of blame to control over the illness course as problematic. First of all, the measures of controllability attributions in some of the prior studies of schizophrenia (e.g., Weisman et al, 1998, and Brewin et al., 1991) are quite different (based on coders’ ratings of statements generated from the Camberwell Family Interview). The authors would be on stronger grounds if the presumed difference was observed across the illness course using similar methods. Second, I find the cognition-emotion distinction muddled in this speculation, and to some extent throughout the paper. Weiner’s attribution of responsibility model, which underlies some of the attribution-EE literature, makes a clear distinction between the appraiser’s cognitive construal and the subsequent affective reaction. To refer to emotional negative representation of illness as “the only attribution that encompasses both cognitive and emotional components (e.g., sense of fear, frustration, anger, worry) blurs the conceptual distinction between cognition and affect.

We concur with the Reviewer that our speculation about the possibility of a shift from attributions of blame to attributions of control over the illness course would be more grounded if there were previous findings supporting this notion that used similar methods to ours. However, please note that we make it very clear that this idea is offered as a speculation: “It is thus attractive to speculate that attributions of blame toward the patient may be emotionally-driven in response to the crisis situation defining the early stages of psychosis, but that in later and chronic stages they may turn into stronger beliefs of patient’s personal control over the illness”; page 16, lines 481 to 484). To our knowledge, there is only study measuring the change and/or stability of relatives’ illness attributions over time in early psychosis using the same method for the assessment of relatives’ attributions (Barrowclough et al., 2014). Specifically, this study found that most relatives’ illness attributions (including blame and control toward the patient) were stable over a period of 6 months in a sample of first episode of psychosis patients. 

The cognition-emotion distinction presents a very interesting and very challenging issue. Indeed, Weiner’s attributional model (Weiner, 1986) assumes that attributions lead to emotions and this theory greatly influenced the development of the attributional model of EE (Barrowclough & Hooley, 2003). Nevertheless, it is important to note that the conceptualization of the attributional model of EE (Barrowclough & Hooley, 2003) was based on the subjective experience of schizophrenia caregivers (which could be significantly different from that of early psychosis caregivers). Relatives in at-risk and onset stages of the disorder are exposed for the first time to the early signs of psychosis. These potent signs of threat and biographical disruption trigger the development of cognitive appraisals about the causes of the disorder and, possibly at the same time, raise highly intense affective reactions. In fact, it is well established that early psychosis caregivers often report a heightened risk of psychological distress compared to family members of patients with chronic schizophrenia (Martins & Addington, 2001). Thus, it is likely that in the early stages of psychosis (when caregivers are dealing with abrupt disruptions and a peak of emotional distress), it is more challenging to establish a clear-cut unidirectional arrow from cognition (attributions) to emotions, and this possibly unique and intense emotional moment may be co-contributing to the shaping of attributions. in terms. This is indeed an interesting and unresolved debate, as it has been previously suggested that the attributional model of EE based on schizophrenia samples should be tailored to the developmental specificities of early psychosis (Domínguez-Martínez et al., 2017). Finally, we referred to emotional negative representation of illness as: “the only attribution that encompasses both cognitive and emotional components” because the 9 items encompassed in this subscale tap both cognitive and emotional content (i.e., I get depressed when I think about their mental health problems; When I think about their mental health problems I get upset; Their mental health problems make me feel angry; Their mental health problems do not worry me; Their mental health problems make me feel anxious; Their mental health problems make me feel afraid; Their mental health problems make me feel worthless; I get very frustrated by their mental health problems; I feel a sense of loss due to their mental health problems). 

4. On a few occasions the authors point out that their findings are consistent with their prior work (Domínguez-Martínez et al., 2014; Dominguez-Martinez et al., 2017) and other’s work as well. For example, they wrote on p. 13 that “The significant associations of relatives’ anxiety and depression symptoms with EE dimensions at both baseline and follow-up is consistent with previous cross-sectional findings [14-16, 18]. Also, on page 14 they noted, “Regarding the association between illness attributions and EE, findings indicated that attributions of blame toward the patient were significantly related with EE-criticism at baseline. This result support (sic) previous findings of cross-sectional studies [14,17] and lends further support to the attributional model [22], which states that relatives who believe that patients are guilty of their behaviors are more prone to manifest critical attitudes. What concerns me is that earlier in the paper they report that the findings of the previous two studies were based on 78 relatives and 44 patient-relative dyads from the current study. That turns out to be 86% of the relatives (78 of 91) and 95% of the dyads (44 of 46). To refer to the findings from previous studies based on nearly the same sample as being supportive of the current findings is potentially misleading.

We apologize if we did not clearly describe the difference between previous cross-sectional reports and the current study. Thus, we summarize below the differences between the papers published by Domínguez-Martínez et al. (2014, 2017) and the present manuscript: 

o In the present manuscript we used an extended sample of early psychosis patients and their respective relatives at baseline (i.e., 91 dyads) as compared to our previous cross-sectional studies (n=78 relatives in Domínguez-Martínez et al., 2017; n=44 dyads in Domínguez-Martínez et al., 2014). As mentioned, the study of Domínguez-Martínez et al. (2017) included only 78 relatives (not dyads). In contrast, the present manuscript included 91 dyads at baseline. Thus, the present manuscript not only includes more relatives at baseline (78 instead of 91 relatives), but also an extra sample of 91 early psychosis patients. 

o With all due respect, we think that the Reviewer may have been confused when comparing the dyads included by Domínguez-Martínez et al. (2014) with the dyads included in the present study. Specifically, the reviewer has compared the 44 dyads included in the cross-sectional study of Domínguez-Martinez et al. (2014) with the 46 family members included in the follow-up assessment of the present study. Actually, the cross-sectional study of Domínguez-Martínez et al. (2014) included 44 dyads. In contrast, the present study included 91 dyads at baseline. Therefore, that turns out to be 48% of the dyads (44 instead of 91 dyads), not the 95% of the dyads referred by the Reviewer. 

o Finally, this manuscript presents, for the first time, longitudinal data. We have never reported any data or findings for the 46 relatives who were reassessed regarding EE dimensions at a 6-month follow-up assessment. 

5. The last major point is that the authors refer to the ontogenesis and origin of expressed emotion in the first sentence of the abstract and the last sentence in the first paragraph of the discussion section. Indeed, the researchers are studying these processes early in the illness course. The authors provide no evidence, however, that they captured the actual beginning of the caregivers’ attitudes and emotional reactions that reflect expressed emotion. They might want to be more circumspect in describing the study and interpreting their findings as an examination of the period of early psychosis, which they do throughout most of the paper.

We thank the Reviewer for pointing this out and we completely concur with the Reviewer’s opinion. Following the Reviewer’s observation, we have rewritten the first line of the abstract (page 2, line 43) and the line that the Reviewer has indicated in the discussion (page 13, line 375) in order to be more precise. 

Please note that, we have also replaced the word origins by expression in another sentence of the discussion (page 16, line 466). 

6. It would be helpful to briefly mention the ARMS criteria and the criteria they used to identify first episode psychosis. For example, was there any consideration of time since illness onset as seen in some first episode studies (Srihari et al., 2014)?

Both ARMS and FEP inclusion/exclusion criteria were already reported in the manuscript’s method section (page 6, lines 222 to 230). Following the Reviewer’s recommendation, we have now added the amount of time considered since illness onset to include FEP patients in the study (page 6, lines 226 to 228) as follows: 

“FEP patients met DSM-IV-TR criteria [35] for any psychotic disorder or affective disorder with psychotic symptoms as established by the Structured Clinical Interview for DSM-IV (SCID-I) [36] and presented a first-episode of psychosis within the past two years. Mean duration of illness was 14 months (SD=9.8), although 4 patients slightly exceed the 24-month period (range 1 to 29 months) and 2 reached a length of 33 and 34 months.”

7. In assessing the carers’ level of anxiety, the authors report using SCL-90 R, yet they cite a much earlier paper that refers to the SCL-90 not the SCL-90R.

We thank the Reviewer for pointing this out. It has now been corrected in the revised manuscript. 

8. One limitation that the authors point out is that they did not consider the role of carers’ positive affect. The authors might consider other research that points out that carers’ supportive stance is associated with positive clinical outcomes for early stages of psychosis (Lee et al., 2014).

We appreciate the Reviewer’s suggestion to consider other research examining how family positive attitudes relate with positive clinical outcomes in early psychosis. Following the Reviewer’s recommendation, we have now included the research of Lee et al. (2014) in the revised manuscript (page 17, line 499).

Response to Reviewer 2

1. The present study is examining potential predictors of expressed emotion (EE), specifically emotional over involvement (EOI) and criticism in relatives of early psychosis patients. Overall, this is an interesting study and has important implications for early psychosis interventions. The manuscript is well-written, the justification for the study is clear, and is overall an interesting contribution to the growing field of understanding how family dynamics may interact with an existing vulnerability to severe mental illness. Nevertheless, the methodological concerns stated below should be emphasized. In addition, a few suggestions/comments. 

We thank the Reviewer for his/her positive comments on the manuscript.

2. Introduction. In the first paragraph, please specify or give examples of the "macroenvironmental factors" referenced. Aside from EE, which is the focus of this paper, what else might be at play? Or is this primarily what the authors mean by macroenvironment?

We thank the Reviewer for pointing this out. We have now provided a specific example of macroenvironmental factors (i.e., urbanicity) in the manuscript (page 3, line 72). In answer to the Reviewer’s question, we would like to clarify that we were not referring to EE as a macroenvironmental factor. Actually, EE is conceptualized as a microenvironmental risk factor, as it pertains to the person rather than social level of the environment (e.g., Sheinbaum & Barrantes-Vidal, 2015). We have now specified this in the revised manuscript (page 3, line 73) to make the text clearer.

3. Introduction. At the end of the first paragraph of the introduction, and throughout the paper, be careful not to imply blame on the family system for exacerbating symptoms or causing relapse, but rather clarifying that the family can play an important role in the course of the patient's illness.

We agree with the Reviewer that it is extremely important to clarify this aspect. We have now addressed this issue in the Introduction section (page 3, lines 79 to 82). The text now explicitly states that families are not guilty or are not the primary causal agent of exacerbating symptoms or causing patient’s relapse in the following way: 

 “However, it is important to note that the EE construct does not aim to blame families for contributing unidirectionally to the patient’s clinical worsening. Instead, EE is best regarded as the product of a negative dynamic interaction between patients and their families that can play a central role in the course of the patient’s disorder [6,7].

Since the main goal of the manuscript was to examine the mechanisms underlying the manifestation of EE in early psychosis, rather than analyzing the effect of EE on patients’ clinical outcomes, we consider that it is not necessary to include more statements about this topic in the paper (as the Reviewer suggested).We think that we have repeatedly emphasized the critical importance of considering caregivers’ psychological needs in the early stages of psychosis and that we have been careful about not laying the blame on the family system. 

4. Introduction. In the introduction, family therapy is briefly mentioned, and a specific study (O'Brien et al) is referenced at the very end of the discussion. Although the focus of this paper is not intervention-based, there are important intervention implications. It would be helpful to spend 1-2 lines (earlier than the last paragraph of the discussion) speaking more specifically about how EE is currently addressed in family therapy and how it could be improved with the findings from the current study.

We completely concur with the point made by the Reviewer. We have now included the following text in the Discussion section (page 13, lines 375 to 382): 

“Evidence confirms that early family-based interventions have great potential to reduce high-EE attitudes and improve patients’ outcomes [43,44]. In general, family therapies aimed at reducing high-EE include training in family communication and problem solving in addition to psychoeducation [45,46]. However, relatives’ own needs and the emotional impact of caregiving are still a neglected intervention area in the early stages of psychosis [44]. Thus, these findings highlight that early family interventions would benefit from providing relatives with proper psychological support. This could involve helping relatives to handle harming thoughts and emotions, facilitate emotional processing, and provide specific techniques to reduce negative appraisals and stress. This, in turn, might prevent relatives’ high-EE over the psychotic process.”

5. Methods. It was surprising that there was no inclusion of positive emotion or protective family factors (e.g. warmth, constructive communication) to use as a differential comparison and no mention of these factors until almost the very end of the discussion. These seems like a major limitation of the study (as is briefly mentioned) and should perhaps be addressed earlier on as to why they were not included. For example, I would be interested to know not only whether higher levels of distress and attributions of illness control predict EE but whether, differentially, lower levels of warmth and constructive communication also predict this. And, conversely, how levels of distress and illness attributions differentially predict factors like warmth and constructive communication. This double dissociation would be crucial to improving the impact of this finding.

We completely concur with the point made by the Reviewer about the huge interest of including protective family factors. Indeed, the assessment of positive family environment variables would have provided a complementary, enriched understanding of the research question. However, as mentioned in the limitations of the study, the only available validated measure to examine relatives’ EE [i.e., Family Questionnaire (FQ; Wiedemann et al., 2002)] at the outset of the study does not assess the positive components of the EE construct (i.e., warmth, positive comments, constructive communication). Please note that these data were collected over a longitudinal study and that, unfortunately, the critical importance of positive/protective variables at the time when this study started was much less realized than what it is now—and, consequently, the number of available measures was also scarce. 

We appreciate the suggestion of addressing why positive family factors were not included earlier in the text. However, we honestly think that it is very clear since we define the goals and hypotheses that the study is exclusively focused on the negative components of the EE construct. Finally, as the Reviewer has indicated in detail (and we also globally suggested in the discussion: page 17, lines 499-501), future follow-up studies will be able to provide relevant information on a range of positive family factors (e.g., warmth, positive comments, constructive communication) that might affect the expression of the negative attitudes of the EE construct (i.e., criticism and emotional over-involvement). 

6. Methods. Why weren't demographic variables such as age and sex included as covariates in the hierarchical regression models? There seems to be some variance at least in age and it would be interesting and important to see whether this interacts with the variables of interest to predict EE.

Please note that we did not include any demographic variables (e.g., sex or age) in our analyses because we did not have any specific a priori hypotheses about these variables. Given the number of analyses performed to address our specific research questions, we limited our analyses only to testing a priori hypotheses. Given that the inclusion of sex or age would have been exploratory and given that the study was not designed to sample and test hypotheses about these variables, we did not believe it would have been appropriate to include them in the analyses.

It is also worth noting that the issue of “adjusting” for confounders presents some conceptual and methodological challenges. As noted, given that we did not have hypotheses about sex or age, it would not be entirely clear why we would adjust for them. Secondly, the practice of examining covariates or “adjusting” for them is not completely unequivocal. Partialling out these variables from our hierarchical linear analyses would indeed test whether significant findings occurred over-and-above the effects of sex or age. However, as noted in the literature, such analyses (partialling or covarying) do not “adjust” or “correct” for third variables (in the sense that it does not equate for them). Rather such analyses examine partialled or residualized effects. However, such partialling can fundamentally change the psychometric properties and the conceptual nature of variables. Such analyses also do not provide any information about whether the effects of interest differ between males and females. This would require examining the interaction of sex and age on outcome measures (over and above the component main effects). Again, we did not have any hypotheses about these relationships and did not design the study to test such effects. Therefore, we felt it was important to avoid conducting unhypothesized, exploratory analyses.

7. Results. In table 1, it would be helpful and more clear for the reader to include 6 month observations side-by-side with the baseline assessments for comparison, in addition to t-tests to compare measures across the two time points.

Please note that, as mentioned in the manuscript, relatives’ EE dimensions (i.e., criticism and EOI) were the only variables reassessed at the 6-month follow-up. The rest of the study variables were only assessed at baseline (i.e., patients’ clinical/functional status, relatives’ distress and relatives’ illness attributions). Thus, we think that it is not necessary to include an extra column in Table 1 for reporting only the descriptive statistics of EE dimensions at follow-up. On the other hand, and following the Reviewer’s recommendation, we have now provided in the text the descriptive statistics (Mean, SD) of the EE values at follow-up (page 8, lines 300 to 301). The comparison of EE levels across the two time points was already informed in the text (page 8, lines 301 to 303) by including the results obtained with the Dependent t-test statistics. 

8. General. It would be helpful for the reader throughout the manuscript to make it more clear which measures refer to the patient, and which refer to the relatives. For example, in the abstract, the first line refers to the "ontogenesis of expressed emotion in the early stages of psychosis" but it could be made more clear that this is EE in relatives, not patients.

Thank you for raising this opportunity to present findings is a more precise way. Following the Reviewer’s recommendation, we have now specified in parts of text that might read ambiguous when we were referring to the relatives' variables in order to avoid any confusion (page 2, line 43; page 5, line 187; page 5, line 190; page 15, line 446) 

 Cited references:

o Barrowclough C, Gooding P, Hartley S, Lee G, Lobban F. Factors associated with distress in relatives of a family member experiencing recent-onset psychosis. J Nerv Ment Dis. 2014;202: 40–46. 

o Barrowclough C, Hooley JM. Attributions and expressed emotion: A review. Clin Psychol Rev. 2003; 23:849–880. 

o Martens L, Addington, J. The psychological well-being of family members of individuals with schizophrenia. Soc Psychiatry Psychiatr Epidemiol. 2001; 36:128–133.

o Hinojosa-Marqués L, Domínguez-Martínez T, Kwapil TR, Barrantes-Vidal N. Ecological Validity of Expressed Emotion in Early Psychosis. Front. Psychiatry. 2019; 10:854. 

o Domínguez-Martínez T, Medina-Pradas C, Kwapil TR, Barrantes-Vidal N. Relatives’ illness attributions mediate the association of expressed emotion with early psychosis symptoms and functioning. Psychiatry Res. 2014; 218:48–53. 

o Domínguez-Martínez T, Medina-Pradas C, Kwapil TR, Barrantes-Vidal N. Relatives’ expressed emotion, distress and attributions in clinical high-risk and recent onset of psychosis. Psychiatry Res. 2017; 247:323–329. 

o Lee G, Barrowclough C, Lobban F. Positive affect in the family environment protects against relapse in first-episode psychosis. Soc Psychiatry Psychiatr Epidemiol. 2014; 49: 367–376. 

o Sheinbaum T, Barrantes-Vidal N. Mechanisms mediating the pathway from environmental adversity to psychosis proneness. In: Mason O, Claridge G. editors. Schizotypy: New dimensions. Oxford: Routledge; 2015. pp. 116-131. 

o Wiedemann G, Rayki O, Feinstein E, Hahlweg K. The Family Questionnaire: Development and validation of a new self-report scale for assessing expressed emotion. 2002;109: 265–279.

---

## [Decision Letter · Decision Letter 1]

26 May 2020

Predictors of criticism and emotional over-involvement in relatives of early psychosis patients

PONE-D-19-32832R1

Dear Dr. Barrantes-Vidal,

We are pleased to inform you that your manuscript has been judged scientifically suitable for publication and will be formally accepted for publication once it complies with all outstanding technical requirements.

With kind regards,

Sinan Guloksuz, M.D., Ph.D.

Academic Editor

PLOS ONE

Additional Editor Comments (optional):

Reviewers' comments:

Reviewer's Responses to Questions

**Comments to the Author**

1. If the authors have adequately addressed your comments raised in a previous round of review and you feel that this manuscript is now acceptable for publication, you may indicate that here to bypass the “Comments to the Author” section, enter your conflict of interest statement in the “Confidential to Editor” section, and submit your "Accept" recommendation.

Reviewer #2: All comments have been addressed

2. Is the manuscript technically sound, and do the data support the conclusions?

Reviewer #2: Yes

3. Has the statistical analysis been performed appropriately and rigorously? 

Reviewer #2: Yes

4. Have the authors made all data underlying the findings in their manuscript fully available?

Reviewer #2: Yes

5. Is the manuscript presented in an intelligible fashion and written in standard English?

Reviewer #2: Yes

6. Review Comments to the Author

Reviewer #2: (No Response)

7. PLOS authors have the option to publish the peer review history of their article (what does this mean?). If published, this will include your full peer review and any attached files.

Reviewer #2: No

---

## [Editor Report · Acceptance letter]

5 Jun 2020

PONE-D-19-32832R1 

Predictors of criticism and emotional over-involvement in relatives of early psychosis patients 

Dear Dr. Barrantes-Vidal:

I'm pleased to inform you that your manuscript has been deemed suitable for publication in PLOS ONE. Congratulations! Your manuscript is now with our production department. 

Kind regards, 

on behalf of

Dr. Sinan Guloksuz 

Academic Editor

PLOS ONE